# Viruses and vectors tied to honey bee colony losses

Zachary S. Lamas[1,2]*, Frank Rinkevich[3], Andrew Garavito[1], Allison Shaulis[1¤a], Dawn Boncristiani[1], Elizabeth Hill[4¤b], Yan Ping Chen[1], Jay D. Evans[1]*

1 USDA-ARS Bee Research Lab, Beltsville, Maryland, United States of America, 2 Department of Biology, University of Maryland, Baltimore County, Baltimore, Maryland, United States of America, 3 USDA-ARS Honey Bee Breeding, Genetics, and Physiology Laboratory, Baton Rouge, Louisiana, United States of America, 4 USDA Office of the Chief Scientist, Washington, D.C., United States of America

¤a Current address University of Georgia, Integrated Life Sciences graduate program, Athens, Georgia, USA
¤b Current address ProjectApisM, Utah, USA
* Zachary.Lamas@usda.gov, Zlamas1@umbc.edu (ZSL); Jay.evans@usda.gov (JDE)

## Abstract

Commercial beekeepers in the US reported severe colony losses early in 2025, as colonies were being staged for their critical role in the almond pollination season in California. Average reported losses since the preceding spring exceeded 60%, with substantial variation among operations. Many colonies were still actively collapsing in January 2025, at which time pooled and individual samples were collected and then screened for levels of 13 known honey bee pathogens and parasites. Acute bee paralysis virus and other known viral pathogens were found at high levels in pooled bee samples from all collapsing apiaries. Nevertheless, viral loads did not differ between healthy colonies and colonies in active collapse. However, individual bees exhibiting shaking behaviors and morbidity showed distinctly higher loads of two strains of deformed wing virus. Differences between these two analyses suggest that direct collections of morbid bees provide a complementary diagnostic for causal viruses, a suggestion supported by inoculation experiments that successfully replicated observed pathologies. Since these viruses are known to be vectored by the parasitic mite *Varroa destructor*, mites from collapsed colonies were in turn screened for resistance to amitraz, a critical miticide used widely by beekeepers, including all beekeepers surveyed in this study. A genetic trait linked with miticide resistance was found in all collected mites, underscoring the urgent need for new control strategies for this parasite. While viruses are a likely end-stage cause of colony death, other stressors such as nutritional stress and agrochemicals may have also played significant roles.

**Data availability statement:** All relevant data are in the manuscript and its supporting information files.

**Funding:** This work is supported in part by the USDA Animal and Plant Health Inspection Service (APHIS) fund (8130-0960; 8130-0990) and USDA Farm Service Agency (FSA) fund (#FSA25IRA0012292). The funders had no role in study design, data collection and analysis, decision to publish, or preparation of the manuscript.".

**Competing interests:** The authors have declared that no competing interests exist.

## Author summary

Honey bee populations face episodic major loss events, putting critical agricultural services at risk. One such event occurred from the middle of 2024 to January 2025. We surveyed six of the largest US commercial beekeeping operations as they converged on the January almond pollination season and screened bees for known pathogens and parasites using broadly sensitive primers. Viral pathogens linked with a mite vector were higher than in prior surveys and were exceptionally high in symptomatic individual bees collected from collapsing colonies. Viral inocula derived from symptomatic bees showed variable levels of pathogenicity in bioassays, including one inoculum that was lethal at a 1/10,000-fold dilution. Mites collected on-site carried a resistance allele towards a ubiquitous chemical mite control, belying the fact that mite vectors are increasingly hard to control. While a viral outbreak seems most parsimonious, additional hypotheses for this and other loss events are discussed.

## Introduction

Honey bees, *Apis mellifera*, are an integral component of agriculture, providing key pollination services for a wide variety of crops. Honey bees and other pollinating bee species are estimated to provide US$20–30 billion in pollination services in the United States [1], and up to $387B globally [2]. In the US, most honey bee colonies are moved by beekeepers at least twice per year to provide pollination services and to generate income from hive products [3]. Multiple stressors impact honey bee health throughout this annual cycle. Key biotic stressors include *Varroa destructor* mites, microsporidian and trypanosomatid gut parasites, and viruses, while pesticides present a common abiotic threat. Coupled with erratic availability of pollen and nectar, and driven by additive and synergistic impacts, these stressors are known to impact both individual and colony health [4].

 Honey bees show resilience towards biotic [5] and abiotic [6] challenges inherent to beekeeping. This resilience includes rapid worker production followed by the production of new queens and colony fission events, a key trait that allows honey bees and beekeepers to rebuild populations following losses [7]. Nevertheless, managed honey bee populations regularly experience extreme local declines, involving the near or total collapse of populations. Termed 'disappearing disease' [8] or 'Colony Collapse Disorder' [9], these sudden losses can be locally devastating. Specifically, after severe loss events surviving colonies are insufficient in number or condition to repopulate an area. This results in intense financial and emotional strain on commercial beekeepers and those dependent on pollination services [7]. Without stable populations, these losses reverberate through intertwined and codependent agricultural communities. Historically, these losses have been highest during the winter, when adult worker bees must persist for months before queens begin laying eggs and

the colony is able to rebuild. More recently, summertime mortality has become a significant component of annual losses [10–12], pushing overall loss rates above 50% in recent years [12].

A suite of positive-strand RNA viruses predominate in managed colonies in the continental United States [13]. Among them, the dicistrovirus Israeli acute paralysis virus was associated with sudden colony losses in some colonies in early 2007 [14], while additional dicistroviruses, Kashmir bee virus and acute bee paralysis virus, were linked to weak colonies in a more extensive survey of this event, albeit inconsistently across regions [15]. While localized honey bee colony losses have been documented throughout the history of beekeeping, recent losses are often linked to the mite *Varroa destructor* (hereafter *Varroa*) [16]. This widespread mite vectors some, but not all, honey bee viruses, and has been a major synergist for deformed wing virus (variants DWV-A and DWV-B) in the viral family Iflaviridae. DWV invariably increases in prevalence and magnitude after the introduction of *Varroa* to an area [17], often followed by high colony mortality rates [18–20].

In January, 2025, commercial beekeepers began reporting severe losses in commercially managed operations. These losses were reported just prior to almond bloom, the largest pollination event in the world, during which more than 1.5 million colonies are staged in the Central Valley of California, USA, for a one-month flowering season [21]. Losses occurred for colonies that had spent the prior year throughout the US and that were stored both outdoors and in massive temperature-controlled 'sheds'. As losses unfolded, it was evident that over 60% of commercial beekeeping colonies had died since the prior summer, representing 1.7 million colonies with an estimated financial impact of US$600 million [22].

Here, we describe analyses based on six large commercial beekeeping operations that experienced these severe losses. The operations represent a collective summertime high of 183,750 managed colonies, roughly 6.8% of all managed colonies in the United States. We provide a detailed descriptive study of colony conditions, morbidities, and operation losses in light of management records and summertime locations. Quantitative analyses of 13 key parasites and pathogens point to RNA viruses as a key companion of morbid bees and commercial colonies. Critically, specific viruses were abundant in both pooled colony samples and in individual bees showing behavioral morbidities. Viruses were further implicated via experimental inoculation experiments that both replicated behavioral symptoms seen in the field and caused rapid mortality of adult bees. As in past loss events [14,15], these viruses belong to a subset of bee viruses vectored by the widespread bee parasitic mite, *Varroa destructor.* Additional analyses of *Varroa* mites showed the universal presence of a genetic marker for resistance to the widely used acaricide amitraz [23] in mites from collapsed colonies, highlighting the difficulties inherent in controlling mite populations. While pesticides and other stressors are likely to be additional factors involved in honey bee mortality, these results suggest that actions to mitigate the impacts of parasitic mites and their associated viruses are critical.

## Materials and methods

### Field inspections and collections

Samples were collected directly onto dry ice as described previously [7]. Symptomatic (morbid) bees dying in front of colony entrances were collected before internal colony inspections. Colonies were given a sample identification number and were scored for strength and observable morbidities. A random sample of adult bees from the center of the brood nest was collected as a pooled sample for parasite, pathogen, and pesticide analysis. Bee brood (larvae and pupae) were visually inspected for signs of disease, and both healthy and symptomatic individuals were collected along with any attached *Varroa* mites. Bee bread (stored pollen-based food), wax, honey, and 'entombed pollen' [24] were also recorded and collected.

### RNA extraction and cDNA synthesis for pathogens

RNA was extracted from both pooled samples and individual bees. A pooled sample of fifty bees from each colony was used for molecular detection of common bee parasites and pathogens using an established protocol: Section 4.3.2 "Bulk

extraction of RNA" from [25]. Briefly, each pooled sample was placed in a disposable RNA extraction bag (Bioreba, Reinach, Switzerland) and a 25 mL aliquot of guanidine thiocyanate lysis buffer was added prior to manual homogenization with a marble rolling pin on a flat board. After homogenization, 620 µl of fully mixed lysed cells and viruses were added to 380 µl acid-phenol followed by phase separation, alcohol washes, and elution. After extraction, 5 µL aliquots of total RNA were treated with DNase I (Invitrogen, Waltham, MA) at 37 °C for 1 hour followed by 75 °C for 10 minutes, as per the manufacturer's instructions. First-strand complementary DNA (cDNA) was generated using the iScript Advanced cDNA Synthesis Kit, following the manufacturer's protocol. The synthesized cDNA was diluted 1:5 with molecular-grade water to provide a template in qPCR reactions.

Individual bees were extracted using a Trizol-chloroform precipitate extraction. A 1 mL aliquot of Trizol was added to each individual bee in a screw cap microcentrifuge tube with 10 µl of 1mm sterilized silica beads. Bees were then agitated in a bead mill homogenizer at 4000 RPM twice for 30 seconds with one minute between agitation cycles to prevent samples from over-heating. A 200 µl aliquot of chloroform was added to each microcentrifuge tube after which tubes were shaken twice for 15 seconds. Samples were allowed to incubate for two minutes at room temperature, and then were centrifuged at 4 °C and 12,000 RPM for 20 minutes. The supernatant was transferred to 1.7ml Eppendorf tubes containing 1ml of molecular grade isopropyl and then incubated at -20 °C overnight. Eppendorf tubes were vortexed briefly and centrifuged as previously described. Isopropyl was removed, and then pellets were washed with 80% molecular-grade ethanol, with 15-minute centrifuge cycles during each wash. Pellets were air-dried and then dissolved in 50 µl of molecular-grade water.

## Pathogen quantification

Quantitative PCR (qPCR) was performed using the SsoAdvanced Universal SYBR Green Supermix (Bio-Rad, Hercules, CA) in a CFX96 Real-Time System thermal cycler (Bio-Rad, Hercules, CA) following all manufacturers' instructions. The cycling parameters were as follows: initial denaturing at 95 °C for 30 seconds, followed by 50 cycles of 95 °C for 5 seconds and 60 °C for 30 seconds. A melt curve was generated for each sample following the completion of the run protocol for amplicon validationTable A in S1 Appendix describes PCR primers for all 13 targeted pathogens and parasites, along with controls. Absolute quantification of target copy number was calculated by creating a standard curve using standards with known concentrations (provided by the National Honey Bee Health Survey [26]), calculating the slope across multiple dilutions, and then calculating concentration in samples (*Tables B and C* in S1 Appendix).

## Inoculum preparation and infectivity

Viral inocula were prepared from individual field-collected specimens. Symptomatic specimens, exhibiting erratic behavior, immobilization, or inability to fly, were collected individually from colony entrances and then saved on dry ice until storage at -80 °C. Viral inocula were prepared by separating viral particles from host material through repeated freeze-thaw cycles. Samples were processed individually by grinding with a sterile pestle in 1% phosphate-buffered solution, followed by freezing at -80 °C. Each sample underwent a total of three freeze-thaw cycles, concluded by bringing the total volume of PBS to 1500 µl. Finally, samples were pressed through a 0.22 µm filter and then stored at -80 °C for long-term storage. Total RNA was extracted from a 50 µl aliquot of inoculum using an RNEASY kit according to the manufacturer's instructions. 5 µl of RNA was used as a template for cDNA preparation using a one-step iScript protocol (Biorad), incorporating DNase I and RNaseOut according to the manufacturer's instructions.

Inoculum infectivity was tested by direct injection into purple-eyed pupae procured from an overwintered colony in a managed apiary at the USDA-ARS BRL, Beltsville, MD. Each inoculum was diluted 10-fold in PBS, and then one µl of diluted inoculum plus seven µl additional PBS were delivered to each pupae using a Word Precision Instruments UMP3 UltraMicropump microinjector. Pupae were incubated at 32 °C and 60% RH. Pupae were collected over three time series: time of injection (Time Zero), 36h post-injection, and 60h post-injection, then were flash-frozen at -80 °C until total RNA extraction.

## Longevity impacts of collected inocula

Inocula were tested for virulence on adult bees in controlled cage trials. A frame of newly emerging bees was collected from a single colony in a managed apiary at the USDA-ARS BRL, Beltsville, MD, and incubated at 32 °C and 60% RH. Bees were allowed to emerge for 24h and then were pooled and sorted into treatment groups. Cohorts of 16 bees were injected with one of three dilutions of the inoculum stock in 1% PBS ($10^{-4}$, $10^{-5}$, or $10^{-6}$ of the original inoculum). Additionally, a $10^{-4}$ aliquot of each experimental inoculum was heat-inactivated at 95 °C for one hour, serving as a control. Inocula reflected a fraction of the original bee source (hereafter bee equivalent, i.e.,$10^{-4}$, was the equivalent of 1/15,000 of lysate harvested from one bee). Sixteen adult bees were injected per group for a total of 64 treated bees for each inoculum. A control group that was not injected (n = 16) served as an additional negative control for damage incurred due to injection. Adult bees were housed in P-cups [27] with a fondant feeder and incubated at 32°C and 60% RH. Bees were inspected two hours post-injection, and then every 12h thereafter for a total of 15 days. Morbidity and mortality were recorded at each inspection. An additional trial, lasting seven days, was carried out to test survivorship of a single, virulent inoculum (Inoculum C) diluted by $10^{-7}$, $10^{-8}$, and $10^{-9}$ from the original concentration. This trial also utilized a heat-inactivated control, PBS, and a negative control (N = 172 bees).

## Statistical analyses

We used baseR and RStudio with the accompanying packages: Vegan, MASS, DescTools, lmerTest, survival, survminer, and GGPlot for statistical analysis and figure creation. Our data from both pooled diagnostic screening (n = 113) and individual bee diagnostic screening (n = 68) failed to meet the conditions for normality. As a result, non-parametric test statistics were utilized. Field descriptions by a single experienced individual were used to define colony status. Pooled bee samples were categorized into groups based on colony strength (Strong, Medium, Weak), and individual bee samples were defined by their corresponding group (morbid or asymptomatic). These groups were used as predictor variables in subsequent analysis. A Permutational Multivariate Analysis of Variance (PERMANOVA) using Euclidean distance and 999 permutations was used to assess variances between and within groups, and across operations using the condition of the colony as a predictor variable. A non-parametric Mann-Whitney U test (Wilcoxon rank-sum test) was utilized to test for significant differences in pathogens detected between strong and weak colonies within an operation. Pathogen abundance and prevalence were calculated using descriptive statistics. The interquartile range (IQR) of pathogen abundance was calculated by subtracting the third quartile (Q3) from the first quartile (Q1). Then, the relationship between the abundance of detected pathogens and pathogen load for each colony was used to calculate the Spearman rank correlation coefficient. For analysis on individual bee specimens, the Mann-Whitney U was used to test to measure significant differences in viral load between symptomatic (dying) and asymptomatic bees. A Linear mixed-effects model using Satterthwaite's method for approximating degrees of freedom for t-tests was fitted for each virus (DWV-A and DWV-B) separately to investigate if source (beekeeper operation) was significantly associated with either virus, using operation as the fixed effect and individual specimens as a random intercept to account for potential non-independence of samples within colonies. Two-way cluster analyses for viral loads in healthy and morbid bees were carried out via hierarchical clustering (SAS-JMP version 18, Cary, NC).

Infectivity tests for viral inocula were analyzed using one-way analyses of variance for viral loads at set timepoints. Survivorship assays were analyzed using a log-rank test to compare survivorship across treatments via Kaplan-Meier survivorship analyses.

## Screening for genes linked to amitraz resistance in honey bees

Individual mites were assayed for a genetic marker linked to amitraz resistance [23]. In total, 39 individual mites, retrieved from 18 colonies across five beekeeping operations, were screened for a robust marker for amitraz resistance (a non-synonymous nucleotide mutation in the gene encoding β2 octopamine receptor, Octβ2R, in *Varroa*). Mites were collected

from both surviving (N = 13) and perished colonies (N = 5). Sixteen mites were collected from hive surfaces in the five recently collapsed colonies, while 23 mites were collected from brood cells in the live colonies. Mites were stored at -80 °C until shipping to the USDA-ARS Honey Bee Breeding Genetics and Physiology Lab in Baton Rouge, where they were again stored at -80 °C until processing. Individual *Varroa* mites were transferred into 2 mL collection microtubes (Qiagen) with a single sterilized 5 mm steel bead and 60 µL of nuclease-free water. Samples were then homogenized during four cycles on a TissueLyser II (Qiagen) at 30 cycles/s for 10s followed by 5-second intervals with rotation of the tubes between cycles. After homogenization, they were centrifuged for 6 minutes at 2272 x g. The supernatant was used immediately for genotypic determination as below.

Allelic discrimination of susceptible (Y215) and resistant alleles (Y215H) of Octβ2R was done utilizing TaqMan technology according to previously published methods [28]. The region of Octβ2R containing the amitraz resistance mutation was amplified using forward (5′-GGA TAC CGT GCT CAG TAA TGCT-3′) and reverse (5′-CTG TCG GGT CGC TTC TAG ATAG-3′) primers (standard oligonucleotides with no modification). With the VIC labeled fluorescent probe (5′-ATG CGC CAA TAA GTG AAT -3′) for the detection of the wild-type allele, and the 6FAM labeled probe (5′-CGC CAA TGA GTG AAT-3′) for detection of the Y215H mutation [28]. Each probe also had a 3′ non-fluorescent quencher and a minor groove binder at the 3′ end.

TaqMan assays were assessed using a Bio-Rad CFX Connect with 10µL reaction volumes comprised of 5 µL of TaqMan mastermix (ThermoFisher), 0.5 µL of TaqMan assay, which includes the labeled probes and primers, and 4.5 µL of DNA extract. Samples were held at 95 °C for 10min, followed by 40 cycles of 95 °C for 10s, then 60 °C for 30s. Wild-type and mutant strains were confirmed with wells containing plasmid DNA. Genotypes were determined using CFX Maestro (BioRad) and exported to an Excel spreadsheet for analysis.

## Results

### Pathogen and parasite levels

We examined pathogen and parasite loads in pooled samples from 69 weak colonies, 5 medium and 39 strong colonies. These samples were tested for 13 different targets (1,243 total tests; Table 1). The median number of detections was 5 for both weak (n = 69) and strong colonies (n = 39), with an IQR of 3 and 4 respectfully). The most prevalent virus in these samples was deformed wing virus A (78%), a longstanding bee pathogen that has surged worldwide due to vectoring by

**Table 1. Prevalence of identified honey bee pathogens and parasites across sampled colonies listed in descending order). The tracheal mite, *Acarapis woodi*, was not detected.**

| Pathogen | Detection (n = 113 colonies) | Prevalence |
|---|---|---|
| Deformed wing virus-A (DWV-A) | 88 | 77.88% |
| Black queen cell virus (BQCV) | 85 | 75.22% |
| Acute bee paralysis virus (ABPV) | 81 | 71.68% |
| Lake Sinai virus (LSV) | 72 | 63.72% |
| *Nosema ceranae* (NOS) | 71 | 62.83% |
| Sacbrood virus (SBV) | 40 | 35.40% |
| *Lotmaria passim* (Lp) | 34 | 30.09% |
| Kashmir bee virus (KBV) | 32 | 28.32% |
| Apis mellifera filamentous virus (AmFV) | 23 | 20.35% |
| Deformed wing virus-B (DWV-B) | 19 | 16.81% |
| Israeli acute bee paralysis virus (IAPV) | 5 | 4.42% |
| Chronic bee paralysis virus (CBPV) | 5 | 4.42% |
| A. woodi | 0 | 0% |

parasitic mites. The dicistrovirus acute bee paralysis virus, also mite-vectored, showed unusually high levels compared to prior surveys (e.g., [26]), and was found in 72% of screened colonies. Surprisingly, while the presence of viruses was notably high in these affected operations, viral load did not differ between pooled samples from weak and strong colonies (Fig 1). Fig A in S1 Appendix shows the wide ranges in viral loads across different colony conditions. Raw data, including no detections, can be accessed through S1 File.

To better assess virus connections to real-time morbidity, we screened symptomatic and asymptomatic bees from individual colonies. Virus levels were strikingly higher in bees showing signs of behavioral impairment indicative of disease or pesticide exposure. In total, 39 symptomatic adult bees were paired with 29 asymptomatic (control) bees. Individual dying bees had significantly higher levels of DWV-A and DWV-B, W = 254, p = .0006, W = 11, p < .0001 (Fig 2). DWV-B was scarce in asymptomatic bees, but was detected in 100% of symptomatic adults, often at high levels (*Table F* in S1 Appendix).

The prevalence and abundance of DWV-A or DWV-B in individual bees did not differ across different beekeeping operations. A linear mixed-effects model revealed no statistically significant impact by operation (F(2, 4.308) = 1.918, p = 0.257 for the DWV-A loads in morbid samples (n = 38). Pairwise comparisons showed no significant differences in DWV-A between Operation 2 and Operation 3 (estimate = 4.311, SE = 4.403, t(2.029) = 0.979, p = 0.430) or between Operation 5 and Operation 3 (estimate = 7.396, SE = 4.127, t(4.292) = 1.792, p = 0.143). A direct comparison of Operation 3 also showed no significant difference between Operation 5 and Operation 2 for DWV-A (estimate = 3.085, SE = 4.979, t(2.337) = 0.620, p = 0.591). In addition, no statistically significant effects were found for DWV-B with respect to Operation overall (F(2, 4.346) = 1.369, p = 0.343), nor through the same respective pairwise comparisons (estimate = 9.297, SE = 4.902, t(2.343) = 1.897, p = 0.179) or (estimate = 7.633, SE = 4.426, t(4.286) = 1.725, p = 0.155) or (estimate = -1.664, SE = 5.500, t(2.605) = -0.302, p = 0.785). A two-way hierarchical clustering analysis (Fig 3) shows that viral patterns differed across morbid bees. While DWV was generally linked to morbidity, and both DWV strains clustered together, some cohorts of morbid bees had high levels of both viruses, while others had high levels of only DWV-B. IAPV and KBV were not detected in any samples, while CBPV notably was detected only in dying bees, albeit infrequently. The bacterial

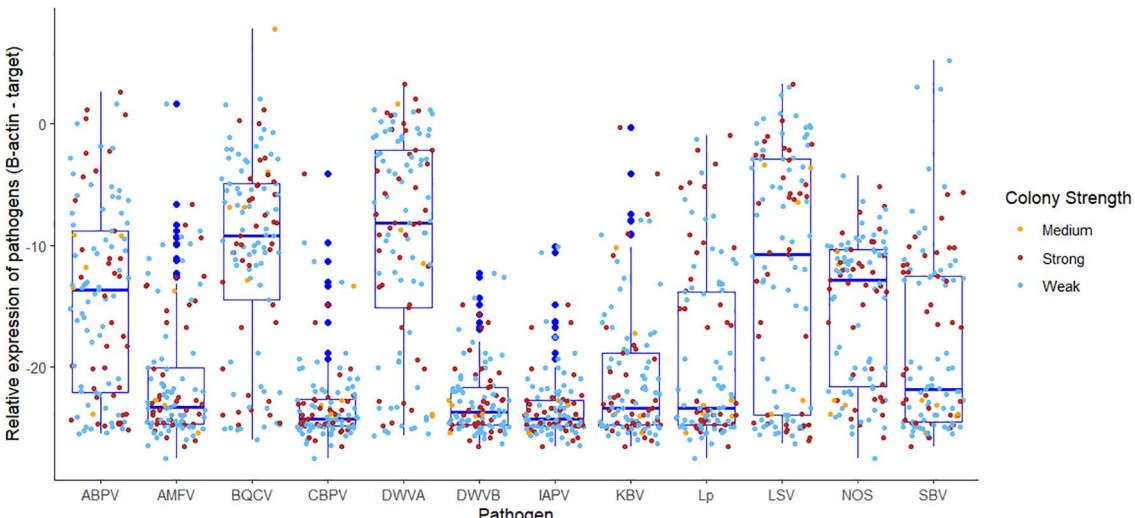

**Fig 1. Diagnostic screening of pooled adult bees from surviving colonies (N = 113).** There was no significant difference in pathogen loads between strong and weak colonies across all colonies in the study (Permanova, F(2,110) = 0.79, $R^2$ = 0.014, p =.663).

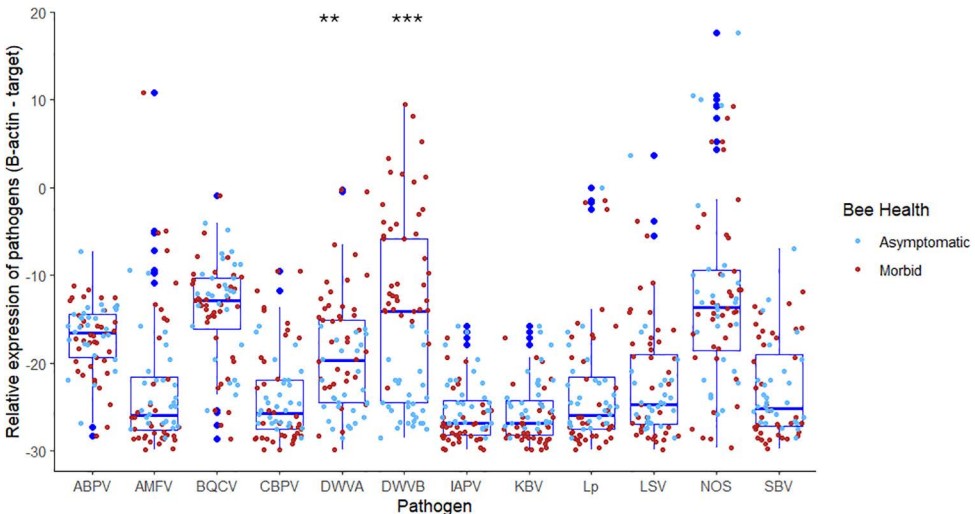

**Fig 2. Diagnostic screening of individual adult bees, showing significantly higher levels of DWV-A and DWV-B in symptomatic individuals.**

pathogens *Melissococcus plutonius* and *Paenibacillus larvae* were not detected in individual bee samples nor was the parasitic mite *Acarapis woodi* (S1 File).

## Infectivity and induced mortality from viral inocula

Viral suspensions purified from individual bees showing behavioral morbidities (inocula A, B, C) contained a range of known bee viruses (*Table D* in S1 Appendix*)*, with inoculum C showing exceptionally high levels of ABPV ($3.52 \times 10^7$ GE/µl). In contrast, the other two did not have detectable levels of ABPV, but instead had high levels of DWV-A and DWV-B (A=$8.43 \times 10^6$ GE/µl, B=$2.27 \times 10^6$ GE/µl). Inoculum D, purified from an asymptomatic bee showing no behavioral morbidities, did not have detectable levels of ABPV, DWV-A, or DWV-B. All three viral pools sourced from morbid bees caused infectivity in injected pupae (*Fig B* in S1 Appendix) at 36 and 60h post-injection (ANOVA, Table 2A), and significantly higher than heat-inactivated controls (ANOVA, Table 2B). DWV-B was amplified in each inoculum sourced from morbid bees. At the same time, ABPV was also robustly amplified in pupae injected with inoculum C. Pupae injected with the inoculum sourced from a single asymptomatic bee, D, showed quantifiable levels of DWV. However, levels at 36 and 60h post-injection were not significantly higher than samples immediately frozen at injection (Time-Zero, $T_0$), indicating that viral amplification did not increase substantially over time.

The single inoculum from an asymptomatic bee neither showed amplification of viral viruses in pupae nor caused significant mortality in adult bees. However, the three inocula sourced from morbid bees all had negative impacts on survivorship when injected into naive bees at a dilution factor of $10^{-4}$ - $10^{-6}$ from the original lysate (Fig 4 and Table F in S1 Appendix). Inoculum C was especially virulent, inducing 100% mortality by 36h in every dilution factor in Experiment 1 ($10^{-4}$ to $10^{-6}$ dilution factor). An additional trial was run, showing that this inoculum significantly affected bee survivorship, inducing 100% mortality in every trial, even when diluted to a factor of $10^{-7}$. When diluted to $10^{-8}$, C induced 44% mortality. While both the mode of injection and maintenance under laboratory conditions are atypical for colonies, this high pathogenicity indicates an extremely virulent biological agent in inoculum C.

## Amitraz resistance in Varroa mites

Of the 39 screened mites, 100% had resistance genotypes in the gene encoding Octβ2R, the known target of the miticide amitraz. These mites were recovered from both perished and surviving colonies (N = 18) from 5 of the

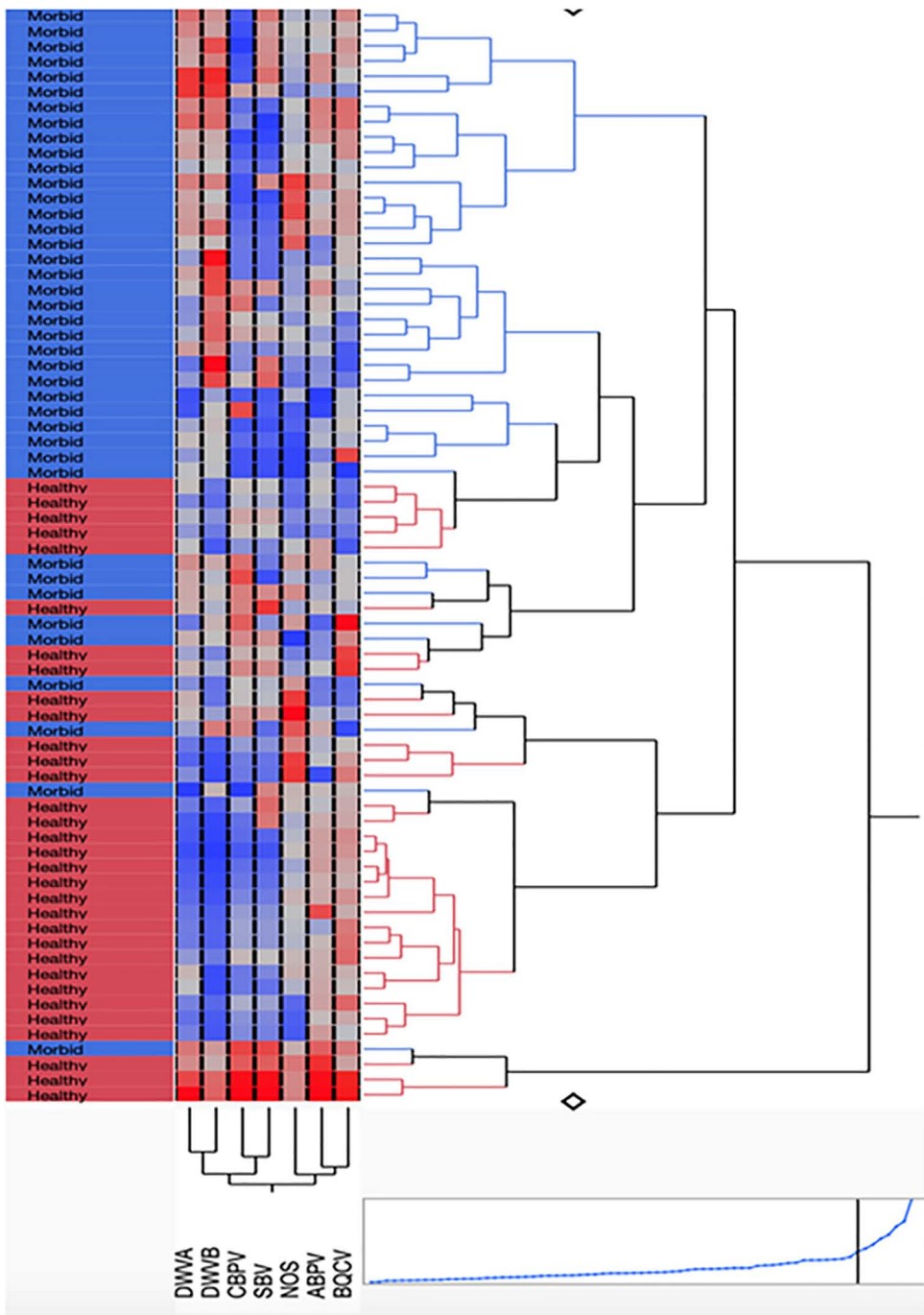

**Fig 3. Two-way cluster analysis showing relative levels (from lowest in dark blue, moderate in muted colors, and highest in dark red) of seven pathogens found in individual bees exhibiting healthy or morbid behaviors.** Two distinct clades with predominantly morbid and healthy bees, respectively, are represented on the top and bottom of the figure.

**Table 2. Infectivity of select viruses found in four inocula 36 and 60h post-injection that were found at higher levels compared to time-zero samples (2A)and compared to viral levels of heat-inactivated controls (2B).**

| Inoculum | Viral Target | Sum of Squares | Df | Mean Square | F | P |
|---|---|---|---|---|---|---|
| Table 2A. Anova table, inoculum infectivity compared to Time-Zero injections. | | | | | | |
| A | DWV-B | 940.8 | 1 | 940.8 | 240.7 | <0.0001 |
| B | DWV-B | 1072.4 | 1 | 1072.4 | 505.5 | <0.0004 |
| C | DWV-B | 280.27 | 1 | 280.27 | 39.26 | 0.0002 |
| C | ABPV | 1795 | 1 | 1795 | 565.1 | <0.0001 |
| D | DWV-B | 100.9 | 1 | 100.87 | 1.987 | 0.201 |
| Table 2B. Anova table, inoculum infectivity compared to heat-inactivated controls. | | | | | | |
| A | DWV-B | 1846 | 1 | 1845.9 | 25.01 | 0.0001 |
| B | DWV-B | 1696 | 1 | 1696 | 20.93 | 0.0004 |
| C | DWV-B | 495.7 | 1 | 495.7 | 17.35 | 0.0006 |
| C | ABPV | 2143 | 1 | 2143.2 | 19.95 | 0.0003 |
| D | DWV-B | 577.5 | 1 | 577.5 | 13.2 | 0.0022 |

6 operations sampled in this body of work. Approximately half (46.8%) of mites screened were recovered from perished colonies. The prevalence of this resistance trait was far lower in the most recent widespread survey of US mites [23].

## Discussion

Sudden losses of worker bees from established honey bee colonies have been noted for decades. Described as 'disappearing disease' [8] or, more recently, 'Colony Collapse Disorder' [9], these sudden events have often remained unsolved, with populations gradually rebuilding in subsequent years [29], in contrast to less extreme loss events that are linked with longstanding threats [12]. In early 2007, widespread colony losses in the United States were heavily scrutinized for biotic causes, with multiple viruses and gut parasites emerging as correlates with weakened colonies. One dicistrovirus, Israeli acute paralysis virus, was linked with colony losses in early samples from this episode [14], while additional viruses were predominant in weak colonies in more extensive surveys [9,15]. Israeli acute paralysis virus and close relatives Kashmir bee virus and acute bee paralysis virus are known to be highly virulent for honey bees and, hence, are frequently screened in honey bee colonies [30–32].

Here, multiple known RNA viruses were quantified in bees collected in the midst of a colony collapse event that impacted a majority of US-managed honey bee populations [22]. High levels were found for two iflaviruses known to cause honey bee mortality [33] and an unusually high incidence was recorded for the dicistrovirus acute bee paralysis virus, compared to prior surveys [26,34]. These three viruses are transmitted between bees via the ubiquitous parasitic mite *Varroa destructor* [35,36]. *Varroa* parasitism is controlled largely by the prolific use of combative miticides [37] and approach that is necessary for honey bee colony survival in most regions [38–42]. When the miticides coumaphos and tau-fluvalinate fell out of use due to acquired resistance, beekeepers in the U.S. became largely reliant upon amitraz due to its low impact on bee health, low cost, and ease of use. Amitraz has been suspected of losing efficacy after decades of heavy use [23,43,44] and our results strengthen this claim. In analyzing mites from collapsed colonies, all screened mites showed a genetic marker, the Y215H mutation in the gene encoding *Varroa* Octβ2R, associated with amitraz resistance. While more surveys are required in order to identify commercial apiaries in which this resistance trait is not present,

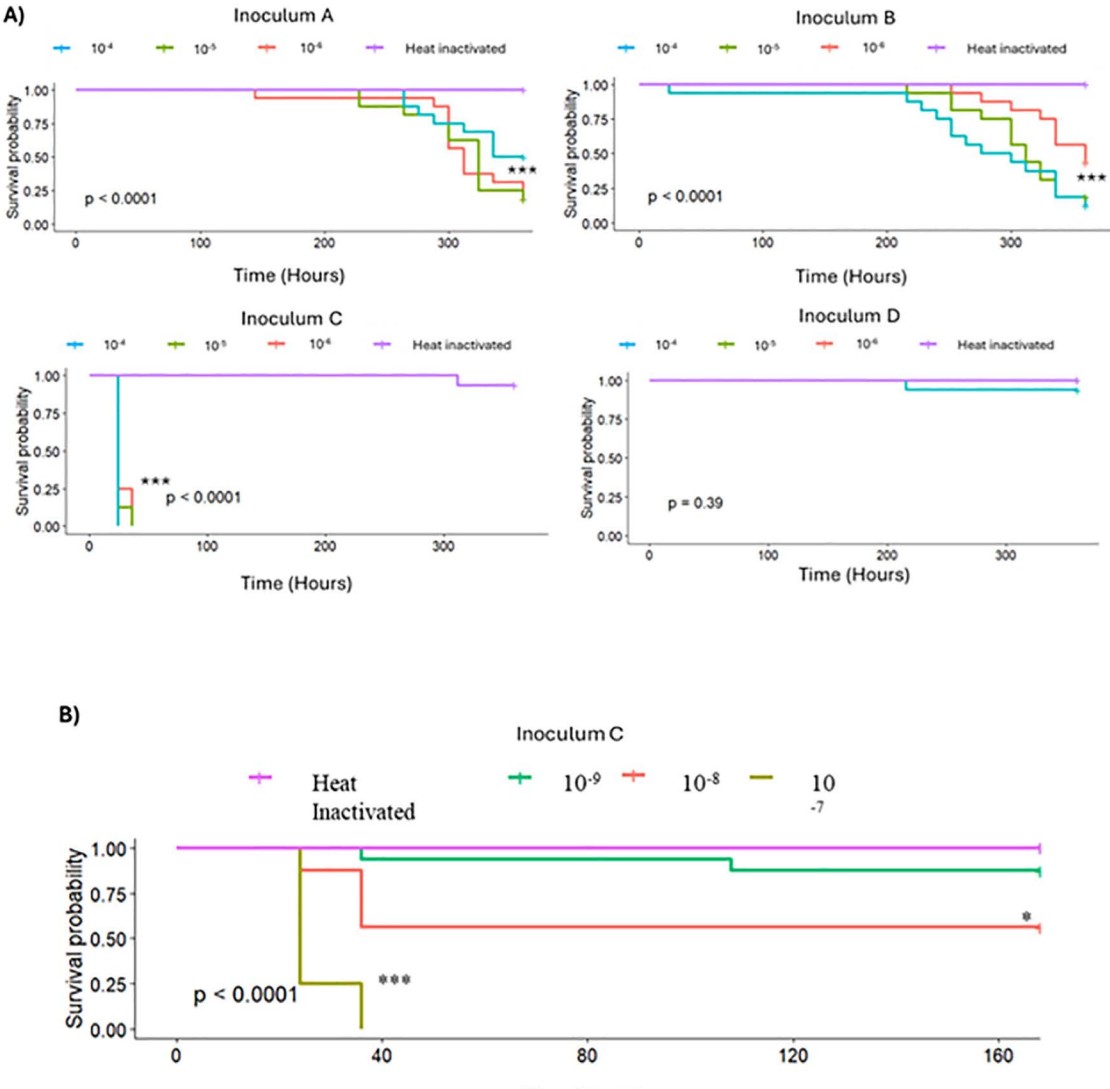

**Fig 4. Survivorship of adult bees in laboratory bioassay.** a) Time-course survival for naive bees inoculated with four inocula sourced from field-collected individual honey bees, diluted to $1.5 \times 10^{-4}$ to $10^{-6}$ from the isolated concentration. In each case, survival curves of bees injected with heat-inactivated inocula of the same source are shown in magenta. b) Survival of bees injected with serial dilutions of inoculum C, from initial concentration to concentrations of $1.5 \times 10^{-7}$, $10^{-8}$, and $10^{-9}$ of the original inoculum.

it is compelling that mites in this survey all showed a resistance trait that was present at only 66% (82/125) in a regional survey of U.S. bees [23]. Beekeepers in our screening were surveyed, and all relied upon amitraz for the majority of their miticide applications, underscoring the challenges faced by beekeepers in controlling mite damage. The universal prevalence of this resistance genotype in screened mites is strongly suggestive that amitraz may have been ineffective at controlling these *Varroa* populations. The removal of amitraz as a viable tool to manage *Varroa* will destabilize beekeeping operations until a replacement miticide or management strategy arrives.

Controlled laboratory assays showed that viral inocula procured from dying bees resulted in productive infection and high pathogenicity. The iflaviruses DWV-A and DWV-B were present in three of these four inocula. One inoculum, labeled C and derived from a single bee showing behavioral signs of disease, had relatively high levels of acute bee paralysis virus alongside DWV-A and DWV-B. This inoculum was especially deadly when injected into naive bees, killing exposed bees even when diluted $10^{-8}$-fold from its original suspension, resulting in a 44% mortality rate for exposed bees.

When coupled with damaging RNA viruses, *Varroa destructor* is the primary threat for managed honey bee colonies [16]. Through their direct feedings on brood and adult bees, *Varroa* act as efficient viral vectors [36,45,46]. *Varroa* actively switch among adult worker bees as they feed [47], allowing typical mite populations to parasitize most members of a populous honey bee colony [46]. Without viruses, these feedings impart little hazard directly to the parasitized bee. However, *Varroa* efficiently vector a suite of honey bee viruses, which subsequently transmit through numerous horizontal routes host-to-host [48]. Virus-infected bees live shorter lives [18]. When the premature death of adult bees surpasses the rate of replacement through brood rearing, colonies are at risk of sudden collapse. Collapsing colonies pose risks to neighboring colonies as they act as a source for *Varroa* and viruses in densely packed operations [49,50].

Cross-sectional surveys have limitations in that prior exposures and specimens lost before sampling result in knowledge gaps. This fact can be partially overcome by carrying out longitudinal studies such that potential causes are recognized before becoming existential threats to colonies [19]. Given the suddenness of these losses, we were not able to collect prior samples. Instead, colonies were sampled directly as they emerged from indoor overwintering sheds or from outside areas with reduced reproduction and activity before the full spring. To refine the search, we also focused on individual bees that were expressing behaviors known to precede death by minutes or hours. The observed discrepancies between pooled colony sampling and targeted sampling highlights the 'survivorship bias' inherent to loss events and arguably helps clarify the challenges faced in pinpointing causes for these events [14,15]. As another factor, colonies often exhibit temporal fluctuations in viral loads, necessitating large sample sizes to identify trends related to colony growth and survival [51]. Coupled with our infection bioassays, we are confident that our results indicate key factors in U.S. honey bee colony declines.

Honey bees suffer from a combination of biotic and abiotic stresses [52], and we cannot rule out the importance of additional causes in these declines. Efforts are ongoing to generate and analyze unbiased RNA and DNA sequencing strategies to test for additional pathogens and potentially identify new pathogens, a successful approach for identifying variants of known pathogens as well as novel species [15,53]. Pesticides are known to exacerbate the impacts of *Varroa* and other disease agents in reducing colony success [54,55] and such synergisms may have compounded the risk to honey bee colonies. Indeed, pesticide exposure alone has been documented to decrease growth and survivorship of honey bee colonies [56,57] and honey bees that forage in heavily agricultural landscapes are always at some risk of pesticide harm. Accordingly, bees and hive substrates are being screened for signs of known agrochemicals.

Nevertheless, this study describes a surprisingly high prevalence of certain bee viruses and a strong correlation between viral loads and native or induced morbidity. These active infections in surviving colonies, coupled with the seasonal increase in *Varroa* levels, present a continued risk this year for managed colonies. Arguably, this supports a near-term shift in mite treatment strategies, while also minimizing additional stressors. Unchecked, mite populations in honey bee colonies grow exponentially, in effect parasitizing a significant fraction of the bee population (Fig 5). A systems approach to protecting and managing honey bees from all threats is needed in order to maintain honey bees as a key player for worldwide agriculture. Commercial beekeepers often face intense financial pressures [3,7], affecting management decisions and the ability to survive chemical and biological threats. Longterm survival of honey bees as a stable agricultural pollinator will require novel protections, from safe and nutritious forage to improved breeding strategies and disease treatments.

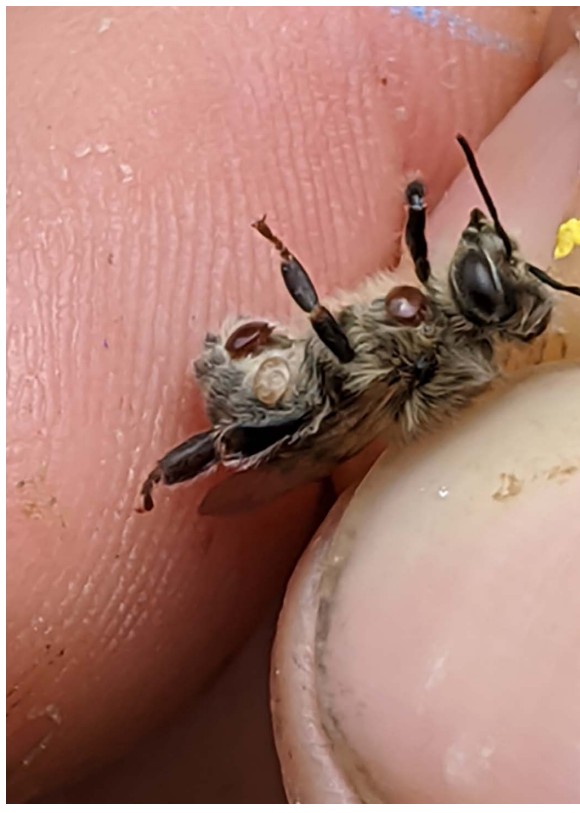

**Fig 5. A heavily parasitized emergent adult honey bee showing female mites that were parasitic during larval and pupal development (Photo Zac Lamas).**

## Supporting information

**S1 Appendix. Including. Table A.** Primer names and associated sequences for qPCR detection and relative quantification of common honey bee pathogens. **Table B.** Primer names and associated sequences for qPCR detection and absolute quantification of four pathogens in experimental inoculum. **Table C.** Dilution factor of four independent inoculum (representing portion of pupa) and copy number per ul of inoculum. **Table D:** B-Actin and viruses screened in inoculums. **Table E:** Prevalence of pathogens found in asymptomatic and symptomatic bees. **Table F:** Inoculums and survivorship of adult bees. **Fig A:** Pathogen detections by colony condition. **A)** Strong, **B)** Medium, **C)** Weak. **Fig B:** Amplification of DWV-B and ABPV targets in individual, injected pupae at time zero, 36 and 60 hours post injection. A subset of pupae were sacrificed at each time point, and used as a representative for viral infection for that specific timepoint. **Fig C.** A typical 'dwindled' colony: honey stores largely intact, queen present, a patch of brood, and adult bee abundance lacking *(Photo Lamas)*. **Fig D.** A typically strong colony, showing coverage across all internal frames, and dense coverage of adult bees on hive surfaces. *(Photo Lamas)*. **Fig E:** A colony which continued to dwindle after uniting with another colony. *(Photo Lamas)*. **Fig F:** A typically "dwindled" unite is picture below. This colony lost approximately 6 frames of adult bee coverage between January 21st, 2025 when it was inspected by the beekeeper, and January 28th, 2025 when inspected by our research team. *(Photo Lamas)*. **Fig G:** Remains of a perished colony includes a mass of recently dead bees and pallet board debris. *(Photo Lamas)*. **Fig H:** An up-close picture of pallet board debris, featuring numerous *Varroa. (Photo Lamas)*. **Fig I:** Immobilized adult worker bees, having recently egressed from their colony. *(Photo Lamas)* (DOCX)

**S1 File. Table of raw data, including Sample, Operation, Colony Strength, and Ct values of pathogen targets and house-keeping gene.**
(XLSX)

## Author contributions

**Conceptualization:** Zachary S Lamas, Andrew Garavito, Elizabeth Hill, Yan Ping Chen, Jay D. Evans.

**Data curation:** Zachary S Lamas, Frank Rinkevich, Andrew Garavito, Allison Shaulis, Dawn Boncristiani.

**Formal analysis:** Zachary S Lamas, Frank Rinkevich, Jay D. Evans.

**Investigation:** Zachary S Lamas, Andrew Garavito.

**Methodology:** Zachary S Lamas, Andrew Garavito, Dawn Boncristiani.

**Project administration:** Dawn Boncristiani, Elizabeth Hill, Yan Ping Chen, Jay D. Evans.

**Supervision:** Yan Ping Chen, Jay D. Evans.

**Validation:** Dawn Boncristiani, Jay D. Evans.

**Visualization:** Zachary S Lamas.

**Writing – original draft:** Zachary S Lamas, Jay D. Evans.

**Writing – review & editing:** Zachary S Lamas, Frank Rinkevich, Dawn Boncristiani, Yan Ping Chen, Jay D. Evans.

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
