## [Decision Letter · Decision Letter 0]

17 Jun 2025

PPATHOGENS-D-25-01256

Viruses and vectors tied to honey bee colony losses

PLOS Pathogens

Dear Dr. Lamas,

Thank you for submitting your manuscript to PLOS Pathogens. After careful consideration, we feel that it has merit but does not fully meet PLOS Pathogens's publication criteria as it currently stands. Therefore, we invite you to submit a revised version of the manuscript that addresses the points raised during the review process.

Please submit your revised manuscript within 60 days Aug 16 2025 11:59PM. If you will need more time than this to complete your revisions, please reply to this message or contact the journal office at plospathogens@plos.org. Please include the following items when submitting your revised manuscript:

We look forward to receiving your revised manuscript.

Kind regards,

Adam G Dolezal

Guest Editor

PLOS Pathogens

Sonja Best

Section Editor

PLOS Pathogens Sumita Bhaduri-McIntosh

Editor-in-Chief

PLOS Pathogens

orcid.org/0000-0003-2946-9497 Michael MalimEditor-in-Chief

PLOS Pathogens

orcid.org/0000-0002-7699-2064

**Additional Editor Comments:**

As the authors can see from the reviewers, the referees felt this was important work and will be impactful, likely receiving significant engagement and scrutiny from beekeepers and the public. However, the current form of the manuscript seems rushed, has some attention to detail issues, and appears to overstate some of the results.

Please address reviewer comments and pay special attention to the following

-lack of varroa mite infestation data and amitraz application data

-comparison to healthy colonies to contextualize the virus loads and justify the argument that the observed levels are higher than ‘normal’

-attention to detail on figures and figures legends; legends are very sparse, figures are hard to read, and there are several inconsistencies between the figures and results.

-The reviewer comments focus primarily on the virus/pathogen aspects of the study, but some touched on the amitraz resistance point and I want to highlight these. For example, the manuscript refers to ‘universal’ presence of a genetic marker associated with miticide resistance. However, the sample size is quite small, the sampling methods are not clear, and no Varroa load or amitraz treatment data is provide, making it very difficult to make such a comprehensive claim.

**Journal Requirements:**

At this stage, the following Authors/Authors require contributions: Zachary S Lamas, Frank Rinkevich, Andrew Garavito, Allison Shaulis, Dawn Boncristiani, Elizabeth Hill, Yan Ping Chen, and Jay Evans. Please ensure that the full contributions of each author are acknowledged in the "Add/Edit/Remove Authors" section of our submission form.

https://journals.plos.org/plospathogens/s/submission-guidelines#loc-parts-of-a-submission

4) We do not publish any copyright or trademark symbols that usually accompany proprietary names, eg ©,  ®, or TM  (e.g. next to drug or reagent names). Therefore please remove all instances of trademark/copyright symbols throughout the text, including:

- ® on pages: 5, and 8

- TM on pages: 4, 5, and 8.

5) Please upload all main figures as separate Figure files in .tif or .eps format. For more information about how to convert and format your figure files please see our guidelines:

6) We have noticed that you have uploaded Supporting Information files, but you have not included a complete list of legends. Please add a full list of legends for your Supporting Information files after the references list.

7) Some material included in your submission may be copyrighted. According to PLOSu2019s copyright policy, authors who use figures or other material (e.g., graphics, clipart, maps) from another author or copyright holder must demonstrate or obtain permission to publish this material under the Creative Commons Attribution 4.0 International (CC BY 4.0) License used by PLOS journals. Please closely review the details of PLOSu2019s copyright requirements here: PLOS Licenses and Copyright. If you need to request permissions from a copyright holder, you may use PLOS's Copyright Content Permission form.

Potential Copyright Issues:

i) Please confirm (a) that you are the photographer of 4, and S1, or (b) provide written permission from the photographer to publish the photo(s) under our CC BY 4.0 license.

8) We note that your Data Availability Statement is currently as follows: "Data is available in accompanying Supplementary File.". Please confirm at this time whether or not your submission contains all raw data required to replicate the results of your study. Authors must share the “minimal data set” for their submission. PLOS defines the minimal data set to consist of the data required to replicate all study findings reported in the article, as well as related metadata and methods (https://journals.plos.org/plosone/s/data-availability#loc-minimal-data-set-definition).

**Reviewers' Comments:**

Reviewer's Responses to Questions

**Part I - Summary**

Reviewer #1: Editors,

The manuscript entitled, “Viruses and vectors tied to honey bee colony losses” by Lamas et. al Evans describes the prevalence and abundance of pathogens (with a focus on viruses)

The authors used pathogen specific PCR to test for 13 pathogens and qPCR to examine the relative quantification of a subset of these pathogens, in honey bee samples collected from several beekeeping operations experiencing high levels of colony losses (including those that are typical of dwindling colonies or more unusual deaths, as nicely described in the supplemental information, though the section on survivorship bias requires editing). The colony level data pathogen levels did not reveal and striking differences in colonies that were weak, medium, and strong (Fig. 1), while the levels in individual morbid vs. asymptomatic bees differed for DWV-B virus (most striking in Fig. 2) and ABPV. In addition, the team determined that Varroa destructor parasites and viral vectors infesting these colonies had a gene/allele associated with miticide resistance (i.e., susceptible or resistant allele of the octopamine receptor (Octβ2R). Overall, the manuscript provides a good description of the data, though it seems some methods, technical aspects and details should be improved and/or clarified prior to publication in PLOS Pathogens. These details may not impact the overall findings, but they would improve the overall quality of the manuscript.

Reviewer #2: This study provides some much-needed empirical data on the current U.S. colony loss situation. The data are valuable; however, in many cases the origins of the samples and other methodological details are insufficiently described. In addition, I have several major concerns that might result in the manuscript being better presented as a survey rather than having confidently (as stated at line 425) identified the causal factor(s) underlying losses. The experimental work on which this causality relies involves testing inocula from just three symptomatic bees on recipient bees from just one colony, which is not sufficient for such a bold claim, in my opinion.

Up to the authors, but I think the discussion would be strengthened if potential socioeconomic factors are described. We are often quick to point to a new disease or poison, but some big-time beekeepers in California have brought up the fact that beekeepers have been feeling the economic squeeze due to labor costs, reduced pollination revenue, etc., which inevitably trickles down to reduced investment in keeping their colonies healthy. I think they are right, that the economic situation shouldn’t be ignored. Just for your consideration.

Reviewer #3: This is an important and time-sensitive paper that provides data on the potential causes of recent high honey bee colony mortality in the USA. In general, it is well-written, although some sections felt rushed, and more attention to detail is required before publication. Given the level of scrutiny and interest it is likely to generate, the authors need to take the time to be precise, flesh out additional required details, and place their results in a broader context. To this end, I have made suggestions (below, in the specific comments section) for the authors to consider.

Reviewer #4: In this study, the authors sampled the many colonies were still actively collapsing in January, 2025, by screening pooled and individual samples for levels of known honey bee pathogens and parasites. The limitation of this study is that they focused not only on known honey bee pathogens and parasites BUT they did not consider the variants that their methods likely missed. At a minimum, this limitation was not discussed at all, especially based on the more advanced technologies available. Both Deformed wing virus strains A and B was identified as possible agents on concern. However, the data as presented does not totally agree with this conclusion (see specific points raised below). Along with DWV, Acute bee paralysis virus, was also found at unusually high levels, either in pooled colony samples or in individual bees exhibiting shaking behaviors and morbidity. The mortality experiments confirm what was previously known about the aggressive and pathological nature of the AKI complex viruses. Only DWV-B could be tested their mortality experiments, and it too show negative effects on bee health (when injected) but not as bad as ABPV.

The "direct collections of morbid bees provide a superior diagnostic for causal viruses" is an overstatement especially given that the morbid bees came form a small subset of teh colonies screen.

**Part II – Major Issues: Key Experiments Required for Acceptance**

Reviewer #1: Points to clarify or address before publication include:

1. Figures 1 and 2 - It may be nice to visualize both of those figures as clustered bar-graphs that separate the designations (e.g., weak, medium, or strong or morbid vs. asymptomatic) to make the data included the means easier to visualize). Including these additional representations of the data as supplemental figures may be best.

2. Fig. 1 – Colony level results, given the natural variability in virus abundance in honey bee colonies the sample sized in this study (i.e. 72 weak and 41 strong colonies) may have been too small to detect differences, unless those differences were large in scale or particular pathogens were only present in weak colonies, etc.. While the authors were limited in the number of samples that could feasibly be obtained during an acute loss event, the authors should comment on this in the Discussion section of the manuscript (see and consider citing - Faurot-Daniels et al PONE 2020, power analysis, virus dynamics).

3. Colony level mite data should be included, if available. If not, a Table indicating what colonies the mites came from and number sampled, etc. should be included in the results or as supplemental material.

4. It would be good for the authors to include more information on miticide resistance (i.e., the percent resistances that is associated the octopamine receptor

susceptible or resistant allele), though some information is included in the Discussion.

5. Line 282-283 and in relevant section of Discussion, since the authors only distinguish DWV strains by qPCR of a relatively small region of the virus genome (~ 130 nt) of a 10,000+ nt genome, it may be that the viruses they are detecting and describing are recombinant strains (i.e., not really DWV-A or DWV-B). Sequencing data would be informative, thought it would only provide a consensus sequence that represents all of the viruses circulating in the bee at the time of sequencing, and therefore, to really distinguish virus strains long read sequencing data would be required. The authors should discuss this in the text.

6. Lines 20-21, and throughout the manuscript the authors refer to usually high levels of virus. To facilitate comparison with other studies, and to ensure accuracy of virus abundance – based on RNA copies, additional sample details (i.e., amount of RNA obtained, virus RNA copies per x ng total RNA) and clarification of the relative qPCR data, which seems to be presented as deltaCt data (i.e., relative to honey bee actin RNA). The authors should cite comparable are cited, D'Alvise et al Ecology and Evol. 2019, and others.

7. While recognizing that it may not be possible to test for all common honey bee pathogens in all samples, it seems it is important for the authors to test for additional pathogens, at least in select pooled samples. Specifically, it would be good for the authors to test for the (1) bacterial pathogens that cause foulbrood diseases (i.e., P. larvae and M. plutonius), while those may be diagnosed in the field, it seems important to also test for them using more sensitive molecular tests; (2) AmFV, a DNA virus that has recently been described as abundant in the US (Cox-Foster- unpublished), or an Iridovirus that has been discussed by beekeepers, etc. and was investigated during previous high colony loss events in the US (e.g. Cornman et al 2012 PONE). I suggest that the authors do additional testing of pooled cDNA samples (e.g., each operation / sample date or healthy vs morbid bees, etc..).

8. Likewise, it would have been good for the authors to do some untargeted analysis (i.e., high throughput sequencing of select pooled samples), which may reveal new pathogens or new/different viral strains that may prevalent in this sample cohort.

9. While sequencing and analyzing pooled samples may be beyond the scope of this article for the sample cohort, the authors should perform sequence analysis on their inoculum. In addition, the description of the inoculum is confusing and should be improved (i.e., “bee equivalent microliters”). As I understand it the authors, made inoculum from bee samples that were ground in liquid nitrogen and subject to several freeze/thaw cycles, and then suspended in buffer and filtered. The team has several inocula described as “CV-3 -4, -5, -7”, which these names may refer to specific samples in the cohort, for the purposes of the manuscript it would be clearer to call them inoculum #1-4, or A, B, C, D.

Table S3 described the copy number of DWV-A, DWV-B, and ABPV in the inoculum as per ul of inoculum, which may be fine - but since the efficiency of RNA extraction from dilute virus stocks etc. varies, and to ensure RNA was adequately extract from all samples, etc. the copy number should be express as per X ng of RNA/cDNA in the qPCR reaction, which is what is measured.

The authors can provide a per lysate copy via converting back to ul lysate, if they know that total amount of RNA isolated from a given amount of lysate. In short, it is critical to this study that the virus inoculum/lysate is well characterized and that the numbers are accurate; as currently presented it is not possible to evaluate this, the inclusion of an excel table including inoculum, RNA quantity, qPCR Ct values, standard curve values, etc. should be included as Supplementary information.

It is also not clear if the data in Table S3 represent six independent isolations of virus RNA from diluted stocks, which were qPCR – which is unlikely give that the numbers are just serial dilutions. What samples were assessed? Maybe undiluted inoculum?

It is not clear what “representing a portion of pupa”, nor “bee equivalents” refers to, it would be best to include a definition and/or description of the equation for “bee equivalents” the authors are referring to. It likely makes sense to their team, and may be accurate, but it may not be clear to other scientists/readers of this article.

To better compare the inoculums, it would be useful for the reader to know the total RNA amount obtained from each inoculum (it would be best to have isolated RNA from 3 dilutions per inoculum to ensure that RNA isolation was uniform, since this essential for absolute quantification), this would also facilitate better comparison of the inoculum.

It is particularly important to know if RNA was sufficiently extracted from inoculum CV-7, if not – it would not be possible to detect pathogens. While inoculation data indicate that this inoculum may be pathogen free, this should be supported by additional data indicating that this wasn’t a failure of RNA extraction (since host RNAs should still be present in these lysates).

PCR or qPCR for host B-actin in the sample samples from which virus quantification data was obtained should be done for all inoculum that were used in this study. If original inoculum is not available, then new aliquots could be assessed if qPCR analyses were done for all viruses and B-actin (i.e., from the same RNA extraction).

10. Table S5 – consider changing “Concentration (Bee Equivalent)” to “Dilution Factor” - or better defining this.

11. Lines 329-332, and Lines 376+ - discussion of results as bee equivalents is not standard and not clear. The authors should improve this, since it is likely they are trying to get across that the pooled bees/pupae that were used to make the virus inoculum had a lot of virus, and in turn could have infected a lot of bees – but without better explanation this is not clear. One virus stock was particularly potent. In Fig. 3, Does the heat-treated inoculum for each trial correspond with the inoculum for each trial (i.e., is it specific)?

12. Table S5 – How many independent experiments of the adult bee survivorship studies were performed? It would be best to have multiple, but since there were multiple cages of the most potent inoculum – that is likely sufficient to distinguish that C5 inoculum is more deadly – since data was not obtained from only one representative cage of bees.

13. Fig. S1 – and other places in the manuscript, it is important that the authors clearly state that T=0, 36, and 60 hrs data are obtained from different animals (i.e., they are not blood draws from the same animals), including additional details in the text, methods, and figure captions would make that clearer. How many pupae per group per timepoint are represented in this figure? Please include a more detailed figure caption.

In addition, the vertical axis labels of this figure appear to be truncated/cut-off in my copy.

14. The representation of the “relative expression of pathogens B-actin-pathogen) using qPCR data requires additional explanation in the methods and figure captions. If I understand correctly, the values on the y-axis are the deltaCt (i.e., B-actin Ct value – pathogen Ct value). If so, it seems the values in Fig. S1 clearly indicate that inoculated pupae have a lot of DWV and ABPV (dCT = 10) compared to those assessed at the zero time point.

Assuming B-actin levels in pupae and adults are similar, the dCT values in Fig. 1 seem to rarely exceed “0”, indicating that the viral copy numbers are much less and don’t often get above the levels of actin mRNA in naturally infected bees.

Example calculation B-actin Ct value =10, high virus Ct value =5, dCT = 5 (high virus)

Example calculation B-actin Ct value =10, virus RNA same as B-actin Ct value =10, dCT = 0 (virus RNA levels similar to B-actin mRNA levels)

Example calculation B-actin Ct value =10, high virus Ct value =15, dCT = -5 (low virus), negative dCT values

If I am understanding Figure 1 correctly, why are there so many very low values for the relative expression of pathogens? Was a sufficient amount of sample assayed to ensure qPCR accuracy?

15. The authors need to include an excel file with the data the used to generate the figures in this manuscript (i.e, sample name, RNA data (if available), qPCR Ct values, etc.), it is difficult to fully evaluate the qPCR data presented in this manuscript without that information.

16. The authors should include details about the linear range of their qPCR assays and their primer efficiency equations.

17. Fig. 2 – The data in this figure is compelling that DWVB levels are strikingly higher in morbid bees compared to asymptomatic bees. How were the bees paired? Did bees have the morphological symptoms associated with DWV infection during development (i.e., crinkled wings, small wings, etc.) or were these just the “generally symptomatic” bees, as described in the paper? Since the abundance of virus in asymptomatic individuals can widely vary, the authors should also show this data as symptomatic individual bee data compared to the “average levels” in asymptomatic bees, and/or using a ranked dCT method, where the largest dCT values from symptomatic bees are paired in a “ranked” manner with the samples that had the largest differences in dCT values for asymptomatic bees, otherwise “pairing” may produce arbitrary results by chance.

18. Tables 2A and 2B – it would be good to better explain values in tables – and include corresponding data as a supplemental excel file. The numbers in the table should be standardized to improve readability (i.e., right justified, a single decimal place).

19. Table S4 and within manuscript, please define the target species for “TRYPS” (i.e., what pathogen(s)), it may be good to include representative target NCBI accession numbers in description (i.e., what species do these primers bind).

20. Line 111-112 Methods

Was RNA isolated from all 25 mL or an aliquot of the total sample? If not, what was the volume of honey bee homogenate used for RNA isolation? What was RNA concentration after RNA isolation and DNase I treatment, since relative abundance calculations are dependent upon standard RNA amounts used for cDNA synthesis (i.e., as opposed to a volume containing an unknown amount of RNA. Unless all qPCR is relative, or based on RNA amounts in RT it is difficult to estimate virus copy number. The same information is required for individual bee samples, or at least the amount of RNA used for RT for each bee sample.

21. Lines 148+. The virus inoculum should be characterized. What viruses (or virus RNAs) were detected in the inoculum preparation? RNA should be extracted from a virus aliquot and relative virus RNA amounts should be quantified. It would be best to sequence the virus inoculum.

22. How did the authors rule-out pre-existing infections in pupae in their infection studies?

23. The description of the association of IAPV with colony losses, could be improved based on current information and further consideration of that study, which compared samples from colony deaths to older/historic samples from other operations, and therefore did not identify accurately identify pathogens associated with CCD-affected colonies (e.g., Lines 413+“Israeli acute paralysis virus, was linked with colony losses in early samples from this loss episode16”; and Lines 57-58). It would be better to highlight findings on IAPV pathogenicity, (e.g., Chen et al PLOS Path 2014) and/or Cornman et al 2012 PLOS One).

Reviewer #2: The infectivity experiments are not sufficiently well-designed on their own, and they would probably be better framed in a different way. Right now, it reads as though they were meant to test if isolates from symptomatic bees cause greater infection than isolates from asymptomatic bees when injected into pupae (a hypothesis that can’t be adequately tested with only 4 isolates). Moreover, the fact that the effect sizes in the infectious vs. heat-inactivated comparisons (Table 2B; which are presumably time-point matched) are so much lower than your comparisons of post-infection vs. time zero virus levels tells me that the injected pupae likely had fairly high levels of these viruses already. Background virus testing is not reported, so presumably it was not conducted. Because of these combined issues, I think these tests would be better framed in one or two lines as a preliminary confirmation that the method of generating inocula yielded infectious substances (i.e. confirming the viruses weren’t inactivated somehow during extraction).

Another question I have is the discrepancy between the methods and the results regarding the number of inocula generated from individual bees (methods say 12, results say 4 – which is it?). If 12, what happened to the others? Why were they excluded?

However, my biggest concern is that the adult survivorship assays were not designed in a way that is sufficient to point to the focal viruses as the cause of the broader U.S. colony losses in question. To achieve this, one would need to show that isolates from diverse sources derived from collapsing colonies cause significant mortality in individuals from a diverse range of colony backgrounds. Currently only four isolates (i.e. extracts from just four individual bees) have been tested on recipient individuals derived from a single colony, which is not sufficient to generalize the results. What if the bees from that colony had particularly poor immunity, or already had high background virus levels?

Finally, the section on amitraz resistance is interesting, but only relevant if it led to higher Varroa levels in the collapsing colonies. Did it? Presumably, these colonies MUST have been sampled to determine Varroa infestation intensity, but these data appear to be missing from the manuscript.

Reviewer #3: The main conclusion is that high levels of three viruses likely contributed to the high mortality of colonies in the operations studied. However, the authors were (understandably) unable to provide pre-death samples. However, they could sample unaffected operations or compare their viral loads and prevalence with data from apparently healthy colonies from other surveys or experiments, or other jurisdictions. They do attempt to provide evidence for their conclusions from infectivity studies, which does help in this regard. While I suspect the authors are correct in that the high virus levels did play a role, the study is weakened by the lack of comparator levels, as well as the fact that they did not rule out other factors such as pesticides or nutrition. In the end, they provide posit that the operations that were affected by high colony mortality had high viruses (although no comparator data are discussed to substantiate this), but this does not demonstrate causation. In fact, their own data from the surviving colonies, which also had high viral loads, could be interpreted as evidence that the viruses did not play a role in the mortality. I strongly suggest that they authors incorporate some form of “control” viral levels, even if all they can provide is evidence from other regions, or historical titres. There must be some quantification of what is ‘normal’ to make the claim that the levels of DWV and ABPV are ‘unusually high’.

Reviewer #4: 1) The distinction and way in which DWV-A & B is reported needs to be consistent and improved. For example, line 239 notes that the “prevalent virus in these samples was Deformed wing virus (78%)” yet Table 1 shows that this is for variant DWV-A. It should be noted the relative absence of DWV-B (16.81%) in this study. Is this expected? Based on previous studies, including the 2024 paper (Hesketh-Best, P.J., Mckeown, D.A., Christmon, K. et al. Dominance of recombinant DWV genomes with changing viral landscapes as revealed in national US honey bee and varroa mite survey. Commun Biol 7, 1623 (2024). https://doi.org/10.1038/s42003-024-07333-9) that focused on the 2021 US survey that DWV-B (including the recombinants) where more prevalent. It should also be discussed the bias regarding the primer designs and how this relates to the DWV recombinants. Moreover, the levels of DWV-A appear to be greater than DWV-B (Figure 1).

2) Figure 1 also indicates that LSV loads were greater than that of ABPV. As presented, the reader is led to infer that the primers used can detect all the variants of LSV. This is not true, right? And its impossible to assess (based on this data) what percentage of variants that are not being amplified or being missed by the primers used.

3) Figure 2 and related tables are not being discussed as it relates to the data in Table 1 & Figure 1. Its notable that both DWV-A & B are associated with morbid bees from these surviving colonies (strong or weak… distinction not made). But considering that 78% of the colonies where DWV-A positive and 73% DWV-B negative, does this not tell us something about the role of DWV-A over -B in the overall role in colony health? Also, please include information if DWV-B (17%) co-occurred with the DWV-A (78%) colonies.

4) As per point #3, Figure 3 is being interpreted that DWV-A & B co-occurred equally across all the morbid & healthy bees found across all the colonies. This was not so, correct? DWV-B was only in 17% of the surviving colonies. These individual bees only came from these 17% colonies. Is this correct?

5) Section “Infectivity and Induced mortality from viral inocula”: please don’t use the term “isolates” when speaking about inocula. Isolates should be used only when referring to individual specimens. In addition, only CV5 (wih the ABPV dominant virus) is described (lines 323-332). The CV3 and CV4 inocula is not described. Line 337 “Figure 3” needs to be “Figure 4”. The mortality curves for these two inocula are not dose dependent and the mortality is far less than CV5.

6) The Amitraz resistance observations… these ae based on only 39 individual mites from 8 colonies. Is the count low because mites were absent? Line 354, “Figure 4” should be “Figure 5”.

**Part III – Minor Issues: Editorial and Data Presentation Modifications**

Reviewer #1: 1. Line 238 – Rephrase this sentence, which read as if 532 different pathogens were detected; suggest

“Samples from 113 colonies were tested for 11 potential pathogens (i.e., 1,243 total tests). There were 532 positive test results . .. . [[describe results in Table 1 more]] (Table 1).

2. Lines 25-26 – maybe rephrase text

3. Line 34 check use of “$” or spacing

4. Line 38 – are other parasitic mites still detected in the US an/or in this sample cohort? If not, consider deleting that clause.

5. Gut parasites – should be updated / more specific for Plos Pathogens (i.e., Nosema ceranae, bacteria . …)

6. Maybe rephrase insecticides are a severe threat.

7. Suggest lower case for all virus names that do not include proper nouns, since italicized names indicate that the specific strain/species has been identified.

See website on how to write virus names - ICTV virus nomenclature

https://talk.ictvonline.org/information/w/faq/386/how-to-write-virus-species-and-other-taxa-names

There are a couple options for writing virus names (e.g., Deformed wing virus-A or deformed wing virus A), using capital letters for everything in the virus name.

8. Line 74 suggest changing “had been lost” to “died” to improve clarity for non-bee experts

9. Figure 3 – the A. and B. labels are missing from this figure.

10. Line 83 suggested changing “were indicated” to “were detected”

11. Consider changing the title to, “Viruses and vectors associated with honey bee colony losses”

Maybe I am too literal, but “tied to” did not seem accurate.

12. In some test/tables “sacbrood virus” mistakenly has a space (“sac brood”).

13. Discussion Lines 425-426 “Coupled with our infection bioassays, we are confident that these results point to a causal factor for a large fraction of honey bee colony losses.” could be revised given that the inoculum used was not completely characterized (i.e., sequenced, tested exhaustively for other pathogens, etc.), but the sentence in the next paragraph addresses this.

14. Discussion Lines 429-430 “sequencing resources for these bees to scout for known and novel parasites and pathogens” suggesting revising to “sequencing libraries (or strategies) to test for additional pathogens and potentially identify new pathogens”.

15. Check placement of $ sign in text

Reviewer #2: Introduction:

Line 34: The $387 billion figure seems way too high. Are you sure this is not the figure for all animal pollination, not just honey bees?

Line 41: I don’t know if it’s fair to say that the honey bees have evolved resilience, so much as beekeepers have selected for those production traits. Perhaps this should be reworded?

Methods:

SI units for time are often not followed (e.g., "hours" when it should be "h"). Other small things like that that will be caught in copyediting, I'm sure

Line 132: Please list all pathogens that were tested here. In addition, were non-virus pathogens considered (e.g. Nosema)? This is clear upon reading the results, but it should be first listed in the methods.

Line 144: Please describe the inoculum/specimen origins in more detail. How many inocula did you produce and test? Where did the source colonies originate from (a map might help if it was across the US)? Was each inocula from a different colony? Reading on to the next paragraph, it seems that the answer to the first question is 12 inocula, but this and other details should be stated in the initial description.

Line 151 and elsewhere: Remember to replace “u” with “µ”

Line 170: Please list the concentrations used.

Line 180: Please revise this section to more clearly describe the sample sizes in each type of analysis, what are the dependent variables in each case, and the predictor variables utilized (including whether it was continuous or categorical, and if categorical, how many levels does it have and what were they).

Line 181: R versions and packages should be accompanied by the appropriate citations.

Line 183: How did you determine this? I ask because it is a common misconception that the data itself must be normal; rather, it is the residual data post-modelling. This is worth checking because if your residual data is in fact normal, redoing the analysis with parametric tests would give you more power to detect differences that were previously missed.

Line 185: What is a “descriptive state” and how was it used in the analysis? This isn’t clear to me

Line 187: A PERMANOVA does not “measure group means,” per se. Perhaps reword.

Line 200: The random factor of “specimen” is probably best modelled as being nested within colony, in this case.

Results:

Line 239: Please specify whether this is A or B.

Table 1: Can you provide some justification for why you zero in on ABPV, DWV-A, and DWV-B as “focal pathogens,” when there are apparently other viruses with much higher prevalence than DWV-A? Also, where did these 113 colonies originate from (maybe a map would help)? Furthermore, in the methods, extraction of RNA was described, but not DNA, which presumably must have been done if Nosema and Trypanosomes are included in the pathogen panel. The methods should be updated to reflect this. Note also that “Nosema ceranae” has been misspelled. Finally, please define your limit of detection (i.e. what level qualifies as detected and non-detected).

Figure 1: Why are relative quantification data shown (relative to beta actin) when absolute quantification was conducted, as described in the methods? Also, it looks like some dots are being displayed twice (once as the unjittered “outlier” data points in blue and once as jittered data points in other colors). It would help to include another figure panel showing the data separated by operation, for the reader to be able to assess the variance across operations (even if there is no significant difference). And how many operations are we talking about? Finally, it is very difficult to parse out the colony strength groups by colored dots … plotting them each with their own boxplot next to each other would be a more effective way to show the data. Same goes for Figure 2.

Line 260: It would be helpful to restate the test used in each case in the results text and/or figure legends. Also, some more punctuation may improve the readability here. It is not mentioned, but presumably a Bonferroni correction was used here too? Also, unless I missed it, somewhere in the methods it would be help to describe exactly how “morbid” and “healthy” were defined.

Line 261: What is a “symptomatic control” – do you mean “asymptomatic”? And how does being detected in 100% of symptomatic adults square with having only been detected in 19 out of 113 colonies?

Line 276: This sentence doesn’t make sense to me. It seems like words and/or punctuation are missing

Figure 3: This figure appears to be a screenshot. Please generate a new figure with appropriate resolution as well as a legend that describes the color gradient. The colors are confusing to me, as it seems to mean different things in different places. Also, please justify why only 7 of the 11 pathogens are shown here. For an unsupervised clustering analysis, one would expect all response variables to be included.

Table 2: The data underlying these analyses would be best presented as a main figure, rather than a supplemental figure. Also, which values are for the 36 h time point and which are for 60 h?

Line 299: The way it’s written, I’m not sure if you’re referring to the inoculum or the injected pupa.

Figure 3: A PBS injection was described in the methods but not shown in the results. Please make these data explicit.

Line 332: This is a very big claim and I think it’s a bit misleading to frame the data in this way (as the contents of one bee will never be directly injected into 66 million bees in nature). Such statements should be made very carefully. In a lab setting, with additional stressors and using the most damaging infection method (injection), it’s not necessarily unusual for virus copy numbers in only the double or triple digits to kill adult bees when injected (IAPV is another common highly virulent agent). Using newly emerged bees in your experiments (i.e. before they have acquired their normal gut microbiome and subsequent immune activation) would also lead to higher than normal apparent virulence.

Discussion:

Line 418: Speaking of longitudinal studies, is there anything you can learn by putting your data in the context of the longitudinal data from the annual US national honey bee disease surveys?

Reviewer #3: Specific Comments:

Introduction:

• Line 53 It is nitpicky, but I feel this manuscript will potentially receive much attention, so I wish to help clarify / improve it as much as possible – I suggest considering a different word than ‘venerable” (deserving respect because of age, high position, or religious or historical importance) as venerable has a positive connotation.

• Lines 61-62 Awkward phrasing. Consider something like “losses… have been especially associated with arrival of the invasive honey bee ectoparasitic mite Varroa destructor in a region.”

• Line 67 Is there a missing space before “often”?

• Line 68 Here and elsewhere, consider “high mortality” rather than losses. The colonies are dead, not lost.

• Lines 68-76 Please provide more detail on how these losses were reported, and to whom, and describe the data. It is a basic assumption of the paper that there was a massive die-off, so this needs to be described in a more fulsome way rather than just referencing the Project Apis m website.

• Line 62, 87 You can abbreviate Varroa to V. destructor after the first mention, or Varroa as in line 66. In fact, you have already said it is a widespread bee parasite, so that can be deleted as well.

• Line 91 Consider “compromised honey bee colony health” rather than “bee declines” which could pertain to populations or species.

• The introduction is well-written, and the authors have taken care in their wording. I appreciate that despite the significance of the mortality, they have not resorted to alarmist wording or unwarranted extrapolations as was seen with CCD.

• However, I would encourage the authors to broaden their perspective beyond the US. What was experienced this winter by beekeepers in other regions (e.g. Canada and Mexico in N. America), and how does this die-off compare in scope or timing to other previous ones globally (e.g. Bailey’s papers on Isle of Wight disease)?

• E.g https://capabees.com/shared/CAPA-Statement-on-Colony-Losses-2023-2024_FV.pdf

Materials and Methods

• Please describe the sample collections in more detail in the current submission, rather than relying only on the referenced paper. How many operations / colonies/ bees, from where, how were they selected etc. Please provide enough detail that a reader can understand the population from which your samples were drawn.

• Lines 96-96 This is not informative: “… all samples were collected individually when not collected as pooled samples”

• Line 99 Please describe how they were scored in more detail. What was the scoring system, what were they scored for, specifically?

• Line 100 How was this randomised?

• Line 101 “brood (larvae and pupae) were” rather than was?

• Line 102 What signs of morbidity?

• Line 103-104 How were the stores evaluated and recorded?

• Please provide more information on the field inspections and collections in this paper. You can somewhat abbreviate if details are available in the other paper, but a reader should be able to get the basic idea without simultaneously reading the other paper.

• I am not qualified to review the molecular methods, so I will leave that to other reviewers.

• Line 144, 146 Viral inocula were

• Line 155 Purple-eyed

• Line 158 Do you mean World Precision Instruments microinjector? What model?

• Throughout – please use the Greek letter mu rather than a u in units to indicate micro

• Line 159 Isn’t this one time series with three time points? Or have I misunderstood?

• Line 164 Viral inocula

• Did you test what viruses were in the inocula?

• Line 165 If the bees were from a single colony, shouldn’t apiary then also be singular? Or was it one colony per apiary (and if so, how many apiaries?)

• Is this what was randomised in line 167 – bees from a single colony per apiary?

• Line 152-178 Please specify what the 4 inocula were? Have you pooled individual bees into the 50 ul aliquots x 4, each of which was delivered at three concentrations? Please clarify.

• Line 209 Please add more detail on where the mites came from and how these particular mites were selected.

• Line 213 …here you have switched to mu…

• I appreciate the urgency to publish this specific paper in a timely manner, however the M&M felt as though it was written in a rush, and lacked detail.

Results

• Line 240 “thanks to” is somewhat colloquial or this context

• Line 239 DWV levels should also be mentioned

• Line 240-241 The fact that ABPV levels were high compared to previous data is significant, but should be in the discussion rather than results (and should receive a more thorough treatment there). The results here should mention what the levels were (or refer to a figure / table).

• Line 242 …the presence of viruses

• Line 243 “..the viral load did not differ” please provide statistics in the test here

• It would be nice if you had a comparison to unaffected operations.

• Line 239-243 Were there sig dif in viral prevalence among viruses, or locations / operations?

• Table 1. What do you mean by focal pathogens?

• Figure 1. The axis fonts are small, even reading at 125%. Consider black boxes to distinguish them more from the blue dots for weak colonies. The figure heading should describe any abbreviations (e.g. ABPV, TRYPS). Describe ‘surviving colonies’ more – e.g. surviving colonies from US beekeeping operations experiencing high winter colony mortality. Explain the dark blue dots (presumably outliers) as well as the line (median?) and whiskers…

• Line 251 Do you mean overall # of pathogens? Or for each specific pathogen?

• Line 253 Do you mean no difference among operations in the load of individual pathogens? For any pathogen? What about prevalence?

• Please explain these statistical results more fully, and clearly, and in the text. Please fully describe the figure.

• R2 (superscript)

• Line 256-257 Please describe this more in the methods.

• Line 261 asymptomatic controls?

• Figure 2 See comments about Figure 1. Describe what the asterisks mean. In the figure heading, isn’t DWV-A also sig?

• Line 272-275 What about operations 1,4,6?

• Line 270 Extra “by” before operation

• Line 276 Do you mean respectively? Either way, I don’t think it is needed here. Do you mean among operations?

• Line 281 Across morbid bees (and I think symptomatic is preferable to morbid) or between morbid and healthy bees?

• Figure 3 Heading Perhaps symptomatic rather than “morbid” behaviors. Again, font size should be increased and more detail provided in the heading.

• Line 289 Do you mention these isolates / inocula identifiers in the methods? And please use consistent terms in the text and table 2.

• Do you describe analysing the viral components of the inocula in the methods (apologies if I just missed this). I was looking in the methods for info about the 4 inocula.

• Table 2 Heading Inocula (plural), heat-inactivated (hyphen)

• Line 331 extra comma after that

• Figure 3 heading Inocula (plural), field-collected (hyphen). There is no A and B on the figure as referenced in the heading?

• Lines 348-349 This is M&M

• Line 351 Far less common (no d)

• Figure 4 is unnecessary

Discussion

• Lines 361-363 To make the case that levels in affected operations are high, you need to compare the prevalence/ levels to either unaffected operations or to other survey/published data. How (quantitatively) do the levels compare?

• Line 363 Make the case that it is unusually high.

• Line 365 Reference that they are transmitted by Varroa?

• Line 366 Which surveys (citation).

• Line 367 Also Bahreini, Rassol, et al. "Arising amitraz and pyrethroids resistance mutations in the ectoparasitic Varroa destructor mite in Canada." Scientific Reports 15.1 (2025): 1587. And Marsky, Ulrike, et al. "Amitraz Resistance in French Varroa Mite Populations—More Complex Than a Single-Nucleotide Polymorphism." Insects 15.6 (2024): 390. As well as other references.

• More detail on these operations and their management is required in the methods. For how long and how often and at what does have they used Amitraz?

• Line 375 Previously you did not have umlauts on naïve (lines 325, 337)

• Line 392 “Viral infected bees live shorter lives.” Requires citation.

• Line 393 “crashing” is overly colloquial

• Lines 396 (Has it been kept in check…?)

• 396-407 This seems like a rabbit hole into acaricides that distracts from the main point of the paper. Many beekeepers in other countries (even commercial operations) do not reply on amitraz, so I would argue that there are already alternatives. Suggest this bit is shortened.

• Line 408-415 I was looking for this in the intro – move up.

• Line 415 There is a comma that should be superscripted

• Line 434 I strongly encourage you to strengthen this case. While you cannot get pre-loss data, can you provide data from operations that were not affected, or at least use quantitative evidence from other recent studies and surveys? What levels of these focal viruses are typically found in healthy colonies at least? We need a contrast to provide evidence that these values are high.

• Line 438- Definitely – see French, Sarah K., et al. "Honey bee stressor networks are complex and dependent on crop and region." Current biology 34.9 (2024): 1893-1903.

• In general, the manuscript felt rushed. More detail is required in many places, and more attention to the details of the text, headings, and formatting is required in numerous places. The discussion could be improved by a more thorough treatment of the literature, especially non-USA literature.

Reviewer #4: Line 169: change “95C” to “95oC”

Line 215: insert space between “30” and “cycles/s”

PLOS authors have the option to publish the peer review history of their article (what does this mean? ). If published, this will include your full peer review and any attached files.

**Do you want your identity to be public for this peer review?** For information about this choice, including consent withdrawal, please see our Privacy Policy .

Reviewer #1: No

Reviewer #2: **Yes:** Leonard Foster

Reviewer #3: No

Reviewer #4: **Yes:** Declan Schroeder

**Figure resubmission:**
---

## [Decision Letter · Decision Letter 1]

4 Nov 2025

PPATHOGENS-D-25-01256R1

Viruses and vectors tied to honey bee colony losses

PLOS Pathogens

Dear Dr. Lamas,

Thank you for submitting your manuscript to PLOS Pathogens. After careful consideration, we feel that it has merit but does not fully meet PLOS Pathogens's publication criteria as it currently stands. Therefore, we invite you to submit a revised version of the manuscript that addresses the points raised during the review process.

We look forward to receiving your revised manuscript.

Kind regards,

Michael Letko, PhD

Section Editor

PLOS Pathogens

Sonja Best

Section Editor

PLOS Pathogens

Sumita Bhaduri-McIntosh

Editor-in-Chief

PLOS Pathogens

orcid.org/0000-0003-2946-9497

Michael Malim

Editor-in-Chief

PLOS Pathogens

orcid.org/0000-0002-7699-2064

**Additional Editor Comments :**

While the reviewers acknowledge the authors' efforts to address the comments, this manuscript still requires additional work. Three reviewers request that the authors take further effort to address all of the comments regarding the abstract, methodologies and discussion. The next re-submission should include a version of the manuscript that only highlights the changes that were made using a straightforward identification scheme, as the track-changes copy submitted here was difficult to follow. It seems comments between the co-authors are left on the track-changes copy, which makes the rebuttal appear rushed and incomplete.

**Journal Requirements:**

1) We have noticed that you have uploaded Supporting Information files, but you have not included a list of legends. Please add a full list of legends for your Supporting Information files after the references list.

2) Please check the labeling of the supplementary tables as two tables are labelled as Table S4.

**Reviewers' Comments:**

Reviewer's Responses to Questions

**Part I - Summary**

Reviewer #1: Editors,

The manuscript entitled, “Viruses and vectors tied to honey bee colony losses” by Lamas et. al has been improved the review process, but requires additional attention, revision, prior to publication in PLOS pathogens. The authors address most of the reviewers’ comments, but there are some that still need to be addressed.

Reviewer #2: (No Response)

Reviewer #3: Summarized in my first review of this manuscript.

Reviewer #4: I commend the authors for their detailed rebuttal and subsequent revisions made to the manuscript. I do however still feel that the revisions made don't fully reflect all the points covered in the response to reviewers. The Discussion section does not go far enough to address the limitations of the study. Responses given in the rebuttal should be included in the Discussion. For example, when understanding the merits of PCR vs metagenomics, it would be appropriate for the authors to acknowledge (in line 362) that the inocula or resultant infections might not have been the result of the wild-type viruses but their recombinants. This is also true when it comes to the PCR results from the sampled colonies (Table1). As per their owns words in the rebuttal: "Applying these techniques [e.g. Hesketh-Best et al... metagenomic study] to individual colony samples or apiaries, when feasible, and connecting with colony status, will be insightful." and "As above, we would love to do that and also, of course, encourage others to collect and compare in this way from similar events."

Its perfectly understandable that time and limited resources led to the choice of approaches used, however, given the resolution of sequencing over PCR as a surveillance tool, especially with the discovery of recombinants in honey bees, this study needs to acknowledge these limitations.

**Part II – Major Issues: Key Experiments Required for Acceptance**

Reviewer #1: Points to clarify or address before publication include:

1. The authors should revise abstract based on previous reviewers’ feedback (i.e., Reviewer #4).

Specifically, authors should either remove claims made in the abstract that symptomatic bees are the best to sample the cause of honey bee colony losses, or include some more discussion about this claim. The affected/symptomatic bees don’t make up a high percentage of the colony. How do the authors know that the viruses associated with these bees are the cause of colony losses? It may be that some rewording in the abstract is needed, and additional clarification in the discussion. This point was made by Reviewer #4, but the abstract was not updated in the re-submitted version.

2. Did the authors indicate from which colonies the mites were obtained? This data would enable coupling of colony virus data with mite information, maybe as a Supplemental Table or Figure (rev Rev #1 point 3).

3. Did the authors see “a strong discrepancy between morbid and healthy bees based on qPCR” between DWV-A and DWV-B? (Reviewer 1 – point not addressed).

4. Characterization of inoculum still requires further clarification in text and/or supplemental table. It is not clear what virus tests were performed. The supplemental table should include all test and a “-“ for below detection levels.

5. Host RNA would pass through a 0.2 uM (200 nM) filter. Re-consider request of Review 1 to use host housekeeping gene.

6. Further correct/clarify, it seems inaccurate to stat that (“1/15,000 of the original bee”), maybe

(“1/15,000 of the original bee lysate”), but not of a bee. The authors should just refer to the “dilutions of virus suspensions” or inoculum dilutions.

7. The authors should avoid the use of “bee equivalents” in the text, although now defined in methods section it is a strange way to refer to inocula, using the dilution factor would be better.

Have you ever read “human” or “mouse” equivalents for virus inoculum in other scientific studies?

8. To better address Review 3’s comment 2, the authors should clearly state, which LSV variants are detected using the qPCR primers utilized in this study.

9. Line 168-169 methods were NOT updated as indicated in response to review. The authors need to include how much of the 25 mL plus 50 bee homogenized sample was used for RNA extraction

10. Excel with RNA data, and qPCR data should be included as supplemental data, as requested by Reviewer 1.

11. Line 276 (and all other instances). As described by Reviewer 1, “titers” is not accurate for the data presented in this study – and should be corrected throughout the manuscript.

12. Line 316 – virus names do not need to be capitalized - See reviewer 1’s comments on virus nomenclature (ICTV) and correct text, as needed.

Reviewer #2: (No Response)

Reviewer #3: Summarized in my first review of this manuscript.

Reviewer #4: No major revisions required except for a Discussion section that covers the limitations of the study design.

**Part III – Minor Issues: Editorial and Data Presentation Modifications**

Reviewer #1: Minor points to clarify or address before publication include:

1. The quality of writing was improved, but could still benefit from editing by a professional scientific writer.

For example, Line 73 – “in the face of” .

In addition, the previous reviewer’s comments were not fully addressed in revised version

2. Other corrections needed, but not limited to, include:

Line 114 – “Deformed wing virus” should not be capitalized

Line 116 and throughout “Varroa” should be “V. destructor”

3. Primer tables should include a column with citations for each primer set.

4. Table S4 and figures – still indicated “TRYPS” when it should be Lotmaria passim (update throughout)

5. Figure S2 – uses “titers” which in not accurate, it is relative RNA abundance. (See previous reviewers’ comments)

6. Supplemental dwindled colony, change “unites” to “combined colonies”

7. Survivorship bias – still has “Figure xxx)”; the authors need to re-review the text and correct any additional mistakes, I did not take the time review it for mistakes.

8. Line 229 remove “by hand”

9. Lines 238-239 “A no-inject negative control was utilized (N = 16) to indicate if damage was incurred by the

shunt injection. incurred damage” , requires revision – maybe “A control group that was not-injected (n=16) served as a negative control for damage incurred due to injection in the mock-infected group (n=XX).”

10. Line 254 (and throughout) – categories “Strong, Medium, and Weak” do not require capitalization.

11. Line 317 define “IQR” in text

12. Line 350 “The source of individual bees (beekeeper operation) did not have a statistically significant

effect on viral targets DWV-A or DWV-B” is an odd way to write this, consider “The prevalence and abundance of DWV-A or DWV-B in individual bees from different beekeeping operations” (or something similar).

13. The use of “viral target” should be revised throughout, it seems the virus name can be substituted in all (or most) instances.

14. Line 371 and/or methods and figure caption should better indicate what viruses the inoculum were tested for; this is NOT indicated in Supplemental Table S3.

15. Line 474-475 - Suggest changing “Israeli acute paralysis virus, like close relatives Kashmir bee virus and Acute bee paralysis virus, remains closely monitored and is known to be highly virulent for honey bees bees” to either

“Israeli acute paralysis virus, which is phylogenetically associated with Kashmir bee virus and Acute bee paralysis virus, is often monitored due to its high virulence in honey bees”. Or Israeli acute paralysis virus, which is phylogenetically associated with Kashmir bee virus and Acute bee paralysis virus, is highly virulent and thus often monitored.

Or – do you want to indicate that all three are monitored? If so, reword.

16. Line 476 should be revised “Here we show a preponderance of RNA viruses in bees collected in the midst of . ..” again is odd wording.

17. Line 497 “. . . led to replicative viruses and high pathogenicity.” Should be revised, maybe “. resulted in productive infection and high pathogenicity.”

18. Line 539 “Varroa destructor is considered the most impactful factor for honey bee colony survivorship. . “ consider revising to “Varroa destructor infestation levels above 3% negatively impact honey bee colony survival, and mites are one of the largest concerns for the beekeeping industry. High mite infestation coupled with RNA virus infections are particularly damaging.” – or some other revision.

19. Line 547 add citations to the following “Virus-infected bees live shorter lives.”

Reviewer #2: (No Response)

Reviewer #3: Summarized in my first review of this manuscript.

Reviewer #4: none

PLOS authors have the option to publish the peer review history of their article (what does this mean? ). If published, this will include your full peer review and any attached files.

**Do you want your identity to be public for this peer review?** For information about this choice, including consent withdrawal, please see our Privacy Policy .

Reviewer #1: No

Reviewer #2: **Yes:** Leonard Foster

Reviewer #3: No

Reviewer #4: **Yes:** Declan Schroeder

**Figure resubmission:**
---

## [Decision Letter · Decision Letter 2]

27 Jan 2026

Dear Dr. Lamas,

We are pleased to inform you that your manuscript 'Viruses and vectors tied to honey bee colony losses' has been provisionally accepted for publication in PLOS Pathogens.

Best regards,

Michael Letko, PhD

Section Editor

PLOS Pathogens

Sonja Best

Section Editor

PLOS Pathogens

Sumita Bhaduri-McIntosh

Editor-in-Chief

PLOS Pathogens

orcid.org/0000-0003-2946-9497

Michael Malim

Editor-in-Chief

PLOS Pathogens

orcid.org/0000-0002-7699-2064

Please make the minor corrections noted by reviewer 1.

Reviewer Comments (if any, and for reference):

Reviewer's Responses to Questions

**Part I - Summary**

Reviewer #1: Editors,

The manuscript entitled, “Viruses and vectors tied to honey bee colony losses” by Lamas et. al has been improved the review process.

A couple points/edits:

Revise Line 140 - “620 ul of lyse cells and viruses” is incorrect since homogenization via rolling pin may not lyse many cells and would certainly not lyse viruses – the lyses happens when the acid-phenol is added. Likely “elution” is not correct either, unless they used a column. Likely the RNA was precipitated and then dissolved.

2. most of the time n= is lower case, but in the Varroa section it is a capital

Reviewer #3: See previous reviews

Reviewer #4: n/a

**Part II – Major Issues: Key Experiments Required for Acceptance**

Reviewer #1: (No Response)

Reviewer #3: None

Reviewer #4: n/a

**Part III – Minor Issues: Editorial and Data Presentation Modifications**

Reviewer #1: (No Response)

Reviewer #3: none

Reviewer #4: n/a

PLOS authors have the option to publish the peer review history of their article (what does this mean? ). If published, this will include your full peer review and any attached files.

**Do you want your identity to be public for this peer review?** For information about this choice, including consent withdrawal, please see our Privacy Policy .

Reviewer #1: No

Reviewer #3: No

Reviewer #4: **Yes:** Declan Schroeder

---

## [Editor Report · Acceptance letter]

Dear Dr. Lamas,

We are delighted to inform you that your manuscript, "

Viruses and vectors tied to honey bee colony losses," has been formally accepted for publication in PLOS Pathogens.

Best regards,

Sumita Bhaduri-McIntosh

Editor-in-Chief

PLOS Pathogens

orcid.org/0000-0003-2946-9497

Michael Malim

Editor-in-Chief

PLOS Pathogens

orcid.org/0000-0002-7699-2064